# Early life factors and their relevance to intima-media thickness of the common carotid artery in early adulthood

**Juliana Nyasordzi**[ORCID][1,2]**, Katharina Penczynski**[1]**, Thomas Remer**[3]**, Anette E. Buyken**[1]*

**1** Department of Sports and Health, Institute of Nutrition, Consumption and Health, Paderborn University, Paderborn, Germany, **2** University of Health and Allied Sciences, Ho, Volta Region, Ghana, **3** DONALD Study Dortmund, Department of Nutrition and Food Sciences (IEL), Nutritional Epidemiology, University of Bonn, Dortmund, Germany

* anette.buyken@uni-paderborn.de

## Abstract

### Background

Early life factors may predispose an offspring to cardiovascular disease in later life; relevance of these associations may extend to "healthy" people in Western populations. We examined the prospective associations between early life factors and adult carotid intima-media thickness (IMT), a surrogate marker of atherosclerosis, in a healthy German population.

### Methods

We studied term participants (n = 265) of the DONALD Study, with bilateral sonographic measurements of IMT (4–8 measurements on both left and right carotid artery) at age 18–40 years and prospectively collected data on early life factors (maternal and paternal age at child birth, birth weight, gestational weight gain and full breastfeeding (>17weeks). Mean IMT values were averaged from mean values of both sides. Associations between early life factors and adult IMT were analyzed using multivariable linear regression models with adjustment for potential confounders.

### Results

Adult mean IMT was 0.56mm, SD 0.03, (range: 0.41 mm-0.78 mm). Maternal age at child birth was of relevance for adult IMT, which was sex specific: Advanced maternal age at child birth was associated with an increased adult IMT among female offspring only (β 0.03, SE 0.009 mm/decade, P = 0.003), this was not affected by adult waist circumference, BMI or blood pressure. Other early life factors were not relevant for IMT levels in males and females.

**Data Availability Statement:** Data from this study are available on request. The DONALD Study is still ongoing and the comparably small sample requires specific precautions to avoid potentially identifying

participant information. This has been imposed by the data protection officer Dr. Jörg Hartmann of the University of Bonn as an official ethical restriction. Requests for data access may be sent to the local data protection coordinator Heinz Rinke, email: rinke@uni-bonn.de at the DONALD Study Dortmund of the University of Bonn.

**Funding:** The DONALD Study is supported by the Ministry of Innovation, Science, Research and Technology of the State of North Rhine-Westphalia, Germany. Juliana Nyasordzi is a PhD candidate co-sponsored by the government of Ghana (Ministry of Education) and the German government (Deutscher Akademischer Austauschdienst German Academic Exchange Service (DAAD). The authors declare no conflict of interest. The funders have no role in the design of the study, collection, analyses and interpretation of the data, the writing of the manuscript, and in the decision to submit the findings for publication.

**Competing interests:** The authors have declared that no competing interests exist.

## Conclusion

This study suggests that advanced maternal age at child birth is of prospective relevance for adult IMT levels in a healthy German population and this association may be of adverse relevance for females only.

## 1. Introduction

Interest in research on early life exposures as possible determinants of disease in later life greatly increased since the discovery of the Barker hypothesis that cardiovascular disease (CVD) has its origins in early life [1]. The atherosclerotic process begins early in life, i.e. many years before cardiovascular complications develop later in life [2]. Fatty streaks which are initial lesions of atherosclerosis have been observed in arteries of children less than a year old which increases with age [3]. Also increases in carotid intima media thickness (IMT) and endothelial dysfunction have been suggested as preliminary indications of atherosclerotic plaque development [4, 5] Similarly, alterations in the IMT have been indicated as a marker of subclinical atherosclerosis [6–8] with a high IMT shown to correlate with CV risk factors [6, 9, 10], and to predict CVD [10, 11].

CVD has been associated with low birthweight mostly in populations exposed to maternal undernutrition during gestation [12, 13]. Though the actual pathophysiological mechanisms underlying the association is not fully clarified, it has been associated with the theory of developmental plasticity, i.e. that the developing offspring's system is plastic and sensitive to the nutritional, hormonal and the metabolic milieu in utero, resulting in various physiological or morphological states due to different conditions during development [14, 15]. A decrease in cell numbers in organs due to changes in the intrauterine environment could also account for CVD in adulthood [15, 16].

It is hence plausible that a range of exposures in early life may be associated with later CVD and thus also with adult IMT. The relevance of these associations may also extend to "healthy" people in Western populations due to the presence of such exposures in this population. Specifically, four groups of early life factors remain to be addressed in these populations: (i) Birthweight, which reflects fetal in utero environment, a potential predictor for which evidence is now also emerging in relation to IMT [17–19]. (ii) gestational weight gain may give rise to an obesogenic intrauterine environment specifically among overweight or obese mothers [20], however, there is no data on its relevance for later IMT; (iii) maternal age, since women are now giving birth at a more diverse age range, with a tendency towards older age in Western countries; (iv) early postnatal nutrition, specifically breastfeeding, which has been related to IMT, however producing controversial results [21–23].

To address these four sets of hypotheses we examined the prospective relevance of (i) indicators of intrauterine growth such as birth weight, birth weight-for-gestational-age (i.e. adequate, small and large for gestational age), (ii) pregnancy duration and gestational weight gain, (iii) parental age and (iv) full breastfeeding for adult IMT as a surrogate of CVD among healthy term-born German participants.

## 2. Methods

### 2.1. Study population

The Dortmund Nutritional and Anthropometric Longitudinally Designed (DONALD Study), is a continual, open cohort study undertaken in Dortmund, Germany. Since its commencement in 1985, elaborate records on diet, growth, development, and metabolism has been gathered from over 1,700 children between infancy and adulthood. About 35–40 infants are newly enrolled each year while initial examination commences at the age of 3–6 months, afterwards each child returns for 2–3 more visits in the first year, 2 in the second year and then once yearly until adulthood.

The study protocol conforms to the ethical guidelines of the 1975 Declaration of Helsinki. The study was approved by the Ethics Committee of the University of Bonn. The data protection officer is Dr. Jörg Hartmann and requests for data access may be sent to the local data protection coordinator Heinz Rinke, email: rinke@uni-bonn.de at the University of Bonn. All examinations are performed with written consent of parent and adult participant [24–28].

The children who were initially recruited for the DONALD Study differed considerably in age and prospectively collected data on breastfeeding was not always available. Follow-up into adulthood was not planned at the inception of the study. Since 2004 participants are invited to return for further visits at ages 18, 21, 25, 30, 35 etc. However, not all participants followed this invitation. In addition, due to the open cohort design, many DONALD participants had not yet reached young adulthood by the time of this analysis. IMT measurements are offered to adolescents and adult participants since 2008. In this analysis only IMT measurements in adulthood ($\geq$18 years of age) are used. Mean follow-up until IMT measurement is equivalent to the mean age at IMT-measurement. 607 IMT measurements were available with two persons excluded due to the presence of plaques and stenosis in their measurement. 58 others did not have a minimum of four measurements each on the right and left common carotid artery to be included in the analysis whilst 178 persons were not considered because the images did not fulfill the quality control criteria (see below). Among the remaining 369 persons with acceptable IMT measurements, data from 349 persons were considered who were born term (37–42 weeks of gestation) singletons with a birthweight > = 2500g. A further 84 persons were excluded because they did not fulfill the following minimum requirements: Parents had to have provided information on maternal age at birth, only available for 262, paternal age at birth, only available for 256, birth year, birth weight, gestational weight gain only available for 258 and gestational duration. Hence the sample considered for this analysis includes 265 participants with information on IMT collected between 2009 and 2014. See Fig 1, for the sample size of early life factors and for relevant covariates see Table 1.

### 2.2. Early life exposures

Child birth and maternal characteristics were extracted from the "Mutterpass," a standard document given to all pregnant women in Germany. Gestational duration is calculated according to the mother's last menstrual period.

Maternal weight at first visit at the gynecologist during pregnancy and at the end of pregnancy weight were abstracted from the "Mutterpass," and from these the gestational weight gain was computed.

Birth weight and birth length were recorded at birth. Birth weight-for-gestational-age is defined according to the German sex-specific birth weight and length-for-gestational-age

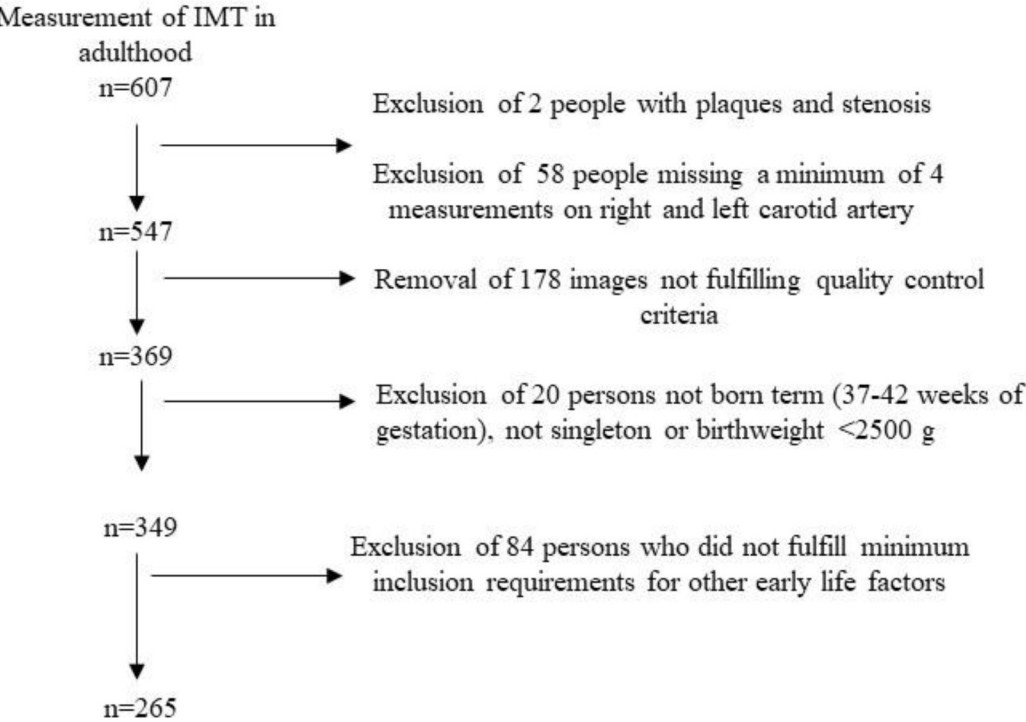

**Fig 1. Participant flowchart diagram for IMT and early life factors.**

curves [29]. Small-for-gestational-age (SGA) is defined as birth weight and length $<10^{th}$ percentile, and large-for-gestational-age (LGA) is defined as birth weight and length $>90^{th}$ percentile. All other infants were classified as appropriate-for gestational age (AGA).

Maternal and paternal age at the time of child birth is assessed at first visit.

Breastfeeding data was assessed upon the child's admission to the study. During first visit either at 3 or 6 months the study pediatrician and/or dietitian enquired from the mothers the duration (in weeks) the infant had been fully breastfed (not given solid foods and no liquids daily except breast milk, tea, or water). If the child is still being fully breastfed, the length of breastfeeding is assessed at successive visits at ages 6, 9, 12 and 18 months until commencement of complementary feeding. The duration of feeding formula or solid foods is also assessed during the visits. A coherent check is conducted on all breast feeding information collected such as the recording of breast milk in 3-day dietary records and information acquired by the dietitians before analysis to minimize errors. From this information the duration (in weeks) of full breastfeeding is calculated [30].

## 2.3. Early adulthood outcome variable: Intima media thickness

Vascular conditions of the left and right common carotid artery (CA) were studied ultrasonographically using high-resolution technology. The Mindray DP3300, tragbares portable digital system was used for this study. The participants were measured in a supine position, head slightly to the right or left after having rested for 10 min. The start point of the measurement was at the beginning of the bifurcation at the left edge of the image with a horizontal vessel course. IMT was measured at 4 points 1 cm before the carotid bifurcation. Images were always taken in the systole. Two images were first taken each on the right and left CA on the

**Table 1. Characteristics of participants in early life and young adulthood.**

| Variables | N | Males | N | Females |
|---|---|---|---|---|
| **Measurements in young adulthood** | | | | |
| Average IMT (mm)[1] | 120 | 0.57 (0.06) | 145 | 0.55 (0.05) |
| Age at IMT measurement (yrs) | 120 | 23.3 (5.7) | 145 | 23.9 (5.1) |
| Waist circumference (cm) | 120 | 83.6 (8.3) | 145 | 75.3 (8.3) |
| BMI at IMT measurement (kg/m$^2$) | 104 | 24.2 (3.3) | 142 | 23 (4.2) |
| Systolic blood pressure (mm Hg) | 103 | 121.3 (10.7) | 140 | 110.5 (9.8) |
| Diastolic blood pressure (mm Hg) | 103 | 75.9 (9.4) | 140 | 72.8 (8.1) |
| Participation in sport (yes/no) [2] | 91 | 88 (96.7%) | 123 | 118 (96.0%) |
| Energy expenditure (kcal) [3] | 55 | 641 (263) | 68 | 363 (260) |
| **Early life factors** | | | | |
| Maternal age at child birth (yrs) | 120 | 30.6 (4.2) | 142 | 30.3 (4.2) |
| Paternal age at child birth (yrs) | 119 | 33.3 (5.1) | 137 | 33.2 (5.0) |
| Pregnancy duration (wks) | 120 | 40 (40, 41) | 145 | 40 (39, 41) |
| Gestational weight gain (kg) | 117 | 12.9 (4.0) | 141 | 12.9 (3.6) |
| High gestational weight gain n (%)[4] | 117 | 20 (17.1%) | 141 | 20 (14.2%) |
| Birthweight (g) | 120 | 3583 (443) | 145 | 3411 (437) |
| Birth weight < 3000 (g) | | 5 (4.2%) | | 28 (19.3%) |
| Birth weight ≥ 3000 to ≤ 4000 (g) | | 93 (77.5%) | | 104 (71.7%) |
| Birth weight > 4000 (g) | | 22 (18.3%) | | 13 (9%) |
| Birth weight by gestation age[5] | 120 | | 145 | |
| SGA | | 12 (10%) | | 16 (11.0%) |
| AGA | | 92 (76.7%) | | 111 (76.6%) |
| LGA | | 16 (13.3%) | | 18 (12.4%) |
| Full breastfeeding | 104 | | 130 | |
| Never (0–2 weeks) | | 29 (27.8%) | | 37 (28.4%) |
| Short duration (3–17 weeks) | | 40 (38.5%) | | 49 (37.7%) |
| Long duration (>17 weeks) | | 35 (33.7%) | | 44 (33.9%) |
| **Additional potential confounders** | | | | |
| Birth year | 120 | 1989 (1985, 1992) | 145 | 1988 (1985, 1991) |
| Firstborn status (yes/no) | 104 | 58 (55.8%) | 129 | 74 (57.4%) |
| Maternal overweight (yes/no)[6] | 116 | 38 (32.8%) | 141 | 41 (29.1%) |
| High paternal educational status (yes/no)[7] | 119 | 70 (58.8%) | 136 | 73 (53.7%) |
| Smokers in the household (yes/no) | 106 | 37 (34.9%) | 134 | 55 (41.0%) |

Values are presented as means (SD) medians (IQR) or frequencies (percentage).

AGA: appropriate for gestational age, LGA: large for gestational age, SGA: small for gestational age.

AGA, LGA and SGA defined according to German sex-specific birth weight and length-for-gestational-age curves.

[1] Average IMT: mean of intima media thickness (IMT).

[2] Participation in organized or unorganized sport: (yes/no).

[3] Estimated energy expenditure during participation in organized or unorganized sport.

[4] High gestational weight gain: yes (>16kg), no (≤16kg).

[5] Birth weight by gestation age

[6] Maternal overweight: yes (≥ 25kg), no (<25kg).

[7] High educational status: yes (≥12yrs of school attendance), no (<12yrs of school attendance).

participants and the images were frozen. Subsequently, measurements were taken at four measurement points on each image.

Quality control was carried out on all images and only images that met the criteria were used for analysis. The criteria for IMT measurement quality control are based on 1) clear representation of the Intima-Media-Complex of the "far wall "shown as echo-rich/echo-poor uninterrupted line, 2) echo-free imaging of the vessel lumen and clear separation of the intima from the lumen, 3) localization of the image section at the beginning of the bifurcation and 4) horizontal course of the vessel within the image. Individual measurement points were discarded, if they were set incorrectly (i.e. the point for measurement was set below or above the visible IMT lines). Mean IMT values were firstly averaged for the right and left sides (i.e. 4–8 measurements), and then an overall mean was calculated from the two averages.

All measurements were performed by the study physicians. Each year, a quality control of IMT measurement by the physicians is carried out. Coefficient of variation (CV) which considers the precision of the measurements within and between the physicians is computed and from 2009 to 2015, the values are $CV_{intra} = 6.95$ and $CV_{inter} = 3.70$. An acceptable precision is given at a value less than 10.

## 2.4. Potential covariates

Anthropometry of study participants were taken at each visit using standard protocol by trained nurses. The participants are dressed in only underwear and are barefooted. Recumbent length of children until 2 years of age is measured to the nearest 0.1 cm using a Harpenden (UK) stadiometer, whilst standing height is measured in children aged older than 2 years to the nearest 0.1 cm with a digital stadiometer (Harpenden Ltd., Crymych, UK). Body weight is measured to the nearest 100 g using an electronic scale (Seca 753E; Seca Weighing and Measuring Systems, Hamburg, Germany). Waist circumference is measured at the midpoint between the lower rib and iliac crest to the nearest 0.1 cm. The trained nurses who perform the measurements undergo quality control, conducted with healthy young adult volunteers [28]. This same measurement procedure is used to measure anthropometry of parents at regular intervals.

The number of smokers in the household was enquired and from this smoking exposure was assessed. The years of schooling was also enquired and from this a proxy of parental socio-economic status was created. A high educational status is defined as ($\geq$ 12 years of schooling).

## 2.5. Statistical analysis

All statistical analysis was conducted using SAS 9.4. Prospective association between early life factors and IMT during young adulthood were analyzed using multivariable linear regression models. IMT was adjusted for age and sex using the residual method.

To evaluate whether sex modifies the association between early life factors and IMT, an interaction analysis was carried out and if a significant sex difference existed, analysis was carried out separately for men and women. Interaction analysis indicated sex interactions for maternal age at child birth and breastfeeding ($P_{interaction}$ = 0.03 to 0.09).

Initial regression models (A) included IMT as the dependent continuous variable and individual inclusion of an early life predictor as the independent variable, adjusted for age at IMT measurement, sex and the physician measuring IMT.

Next, multivariable adjusted models (B) were constructed considering covariates individually for potential confounding in the models in a hierarchical manner [31]. Covariates which substantially modified the predictor–outcome associations by ($\geq$10%) or significantly predicted the outcome were included in the final multivariable adjusted models.

These early life factors were considered as mutual potential covariates in this model (1) early life factors: birth weight (g) considered as both a continuous and categorical variable (i.e.,

<3000g, $\geq$3000g to $\leq$4000g and >4000g), birth weight-for-gestational-age as a three level categorical variables (AGA, SGA, LGA), gestational age (weeks), maternal and paternal age at child birth (years) were considered as continuous variables, gestational weight gain (kg) as a continuous and categorical variable (i.e.$\leq$16kg and >16kg), breastfeeding for >2 weeks (Yes/ No and >16 weeks (Yes/No), first born status (Yes/No) and birth year regressed on age at IMT measurement as a continuous variable. (2) Socioeconomic factors: paternal school education $\geq$12 years (Yes/No), presence of an overweight parent BMI$\geq$25 kg/m$^2$, (Yes/No), smokers in the household (Yes/No). Sensitivity analyses were conducted in subsamples who had provided either information on participation in sport (Yes/No) (n = 123) or on estimated energy expenditure during participation in sports (n = 68) in early adulthood, so as to account for potential confounding arising from adult physical activity levels.

Finally, four sets of conditional models were constructed adding adult waist circumference, adult BMI or adult systolic or diastolic blood pressure to the models, so as to investigate whether observed associations were partly attributable to these variables in adulthood.

Results from regression analysis are presented as adjusted least-square means (95% confidence interval (CI) by tertiles of the respective predictor while P-value is obtained from models using the predictors as continuous and categorical variables. Significance was determined at a p-value of 0.05.

## 3. Results

Early life and young adulthood characteristics of the participants in this analysis are presented in Table 1 according to sex. The minimum and maximum average IMT ranged from 0.41mm to 0.78mm. The mean age at IMT measurement was 23 years in males and 24 years in females. Participants lost to follow up differed slightly from those included in this analysis: mothers of males and females were younger i.e. 29.4 and 29.8 years, children were born earlier i.e. in 1987 and fewer offspring were fully breastfed for a long duration, i.e. 31% and 27% among male and female offspring (for details see S1 Table).

Paternal age at birth was not related to adult IMT (S2 Table). In multivariable analysis, increased maternal age at child birth was associated with an increased IMT among female offspring during young adulthood (P = 0.003, Fig 2), but not in males (P = 0.2, Fig 3) (S2 Table). These associations were not affected by adjusting for paternal age at birth. In addition, inclusion of adult waist circumference, BMI, systolic or diastolic blood pressure in separate conditional models did not affect the relationships (S3 Table).

Sensitivity analyses in the subsample of females, for whom data on participation in sport (n = 123) or data on estimated energy expenditure during participation in sport (n = 68) was available, additional consideration of these variables did not affect the association of maternal age at birth with IMT (S4 Table).

In multivariable analysis on the relevance of full breastfeeding for adult IMT, there was no association in females (P = 0.1, model A and B, Table 2). In males, there was a trend for an association between full breastfeeding and IMT in young adulthood (P = 0.09, model B), which remained when considering adult waist circumference, BMI, systolic or diastolic blood pressure in separate conditional models (data not shown).

When analyses were repeated in subsamples of males with data on participation in sport (n = 91) or estimated energy expenditure during participation in sport (n = 55) in adulthood, full breastfeeding was no longer associated with adult IMT levels, irrespective of considering adult physical activity (data not shown).

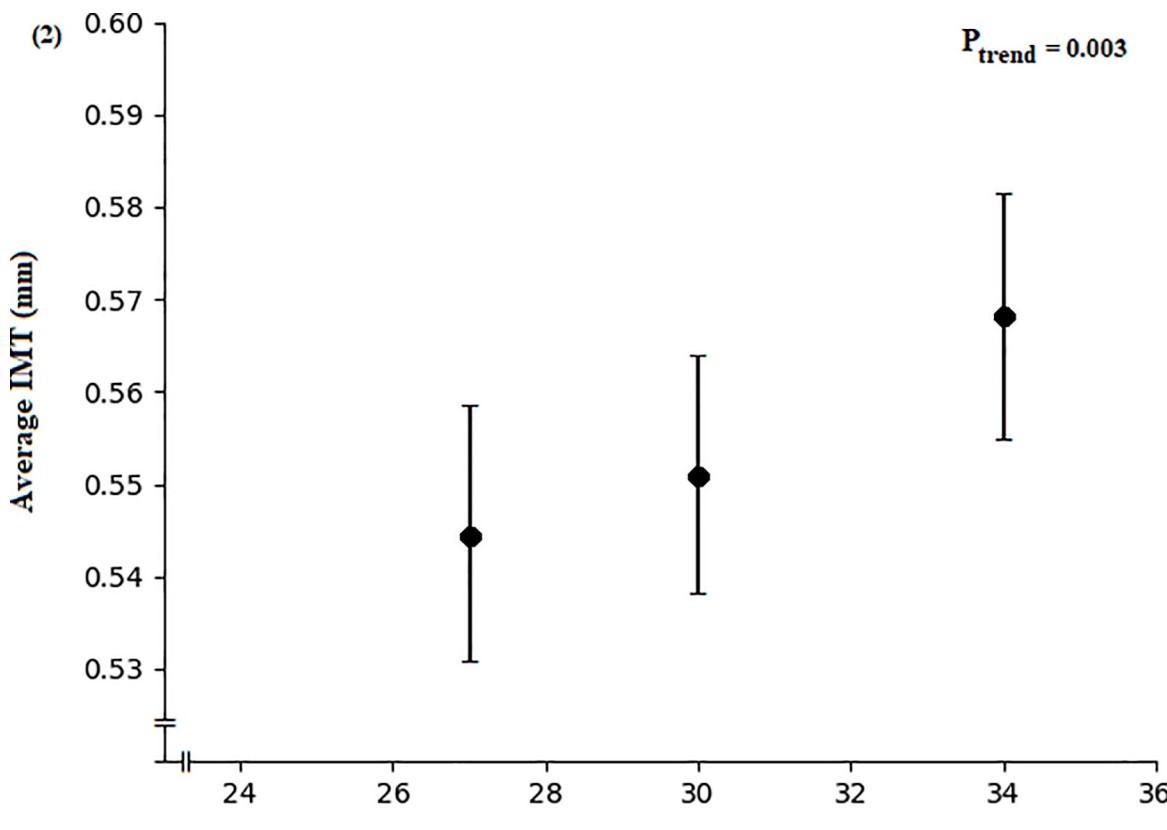

**Fig 2. Association between maternal age at child birth and intima media thickness.** Maternal age at child birth by tertiles of average IMT adjusted for age by the residual method in young adulthood among female participants. Data are means and 95% CI adjusted for adult age at IMT measurement, the physician taking the IMT measurement and birth year (residuals of birth year were calculated on age at IMT measurement), (n = 142).

There was no association of IMT with other early life factors i.e. pregnancy duration, gestational weight gain, birthweight, birthweight according to gestational age (S5 Table and S6 Table).

## 4. Discussion

This study indicates that older maternal age at child birth is associated with an increased IMT of the offspring in young adulthood. Whereas increased maternal age at child birth was associated with an increased IMT in female offspring's, this was not evident in males.

Other early life factors were not associated with IMT in this study. Birthweight and birthweight according to gestational age were not associated with IMT, probably because term infants with mostly adequate birthweight were included in our study.

It is likely advanced maternal age at child birth may programme an offspring for the onset of cardiovascular disease later in life: maternal age at birth was found to be associated with higher infant systolic blood pressure [32] and impaired adult glucose metabolism [33]. In our study, IMT was not associated with paternal educational status (data not shown) and the relevance of maternal age at birth for adult IMT was not influenced by paternal age at birth, hence it is likely the association with maternal age is due to the intra uterine environment rather than socioeconomic factors.

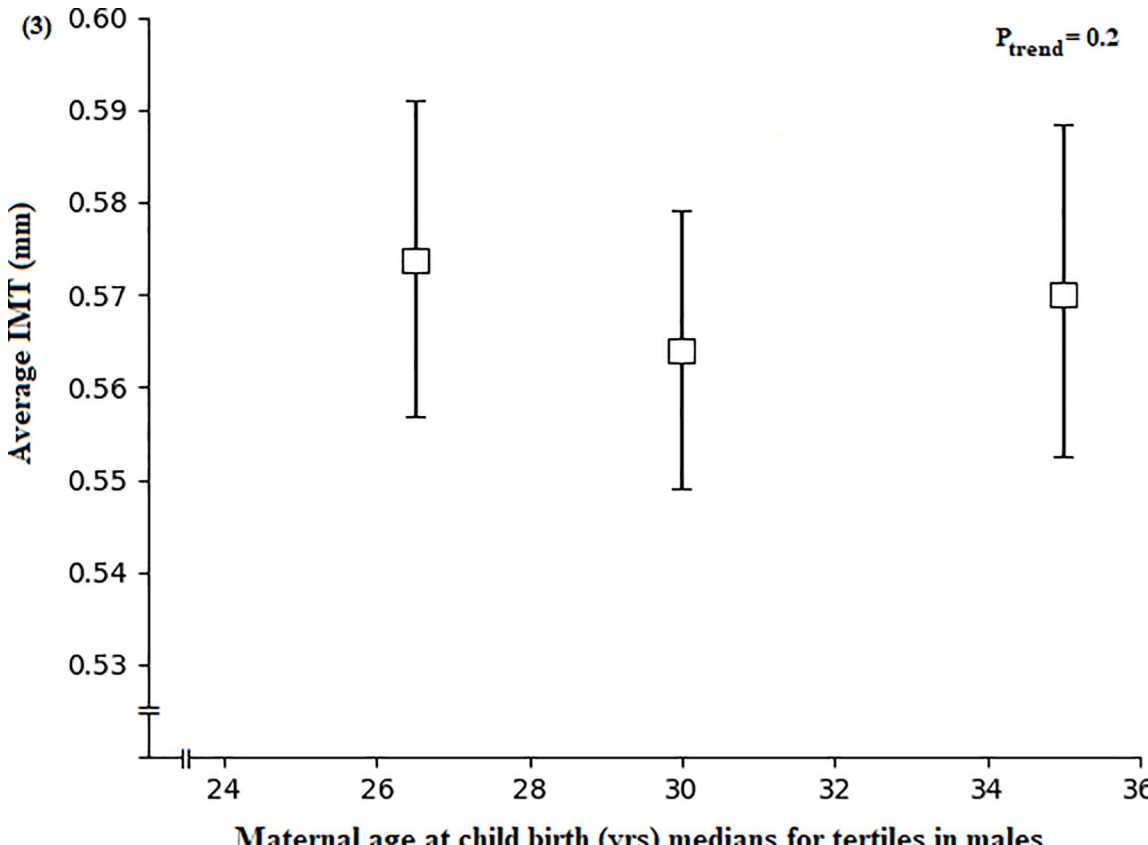

**Fig 3. Association between maternal age at child birth and intima media thickness.** Maternal age at child birth by tertiles of average IMT adjusted for age by the residual method in young adulthood among male participants. Data are means and 95% CI adjusted for adult age at IMT measurement, the physician taking the IMT measurement and birth year (residuals of birth year were calculated on age at IMT measurement), (n = 120).

In line with our study, a study among Chinese participants reported that older maternal age at delivery was adversely associated with IMT among females only [34], with an effect size comparable to ours (ß: of 0.02mm per decade compared to ß: 0.03mm per decade) in our study. Mothers in that cohort were approximately 4 years younger than mothers in our cohort, which may explain the slightly more pronounced effect size in our cohort.

The overall mechanisms between maternal age at child birth and IMT of offspring are not known, yet state of health in mothers advanced in age and placental nutrition could affect fetal development due to variations in uterine artery flow or hormone synthesis in comparison to younger mothers. Likewise, mothers advanced in age are prone to higher levels of CVD risk factors such as higher blood pressure, dyslipidemia or increased oxidative stress [34, 35]. A study using a rat model of advanced maternal age, reported that pregnancy complications were partly due to development of maternal hypertension and altered vascular function in the aged female rats [36]. Hence, both the study in humans and animals suggest that advanced maternal age at child birth may programme an offspring for cardiovascular disease later in life.

In terms of the sex-specific nature of our results it should firstly be noted that men and women have similar CVD risk factors, however, there are considerable variations in the first manifestation and in clinical signs [37]. There is evidence suggesting that IMT among females are more vulnerable to metabolic disorder: Insulin resistance is related to IMT and atherosclerosis solely among females [34], and blood glucose and triglycerides levels associate strongly

**Table 2. Association of full breastfeeding categories and IMT in young adulthood among females and males.**

| | N | Average IMT (mm) | | P trend |
|---|---|---|---|---|
| | | **Full breastfeeding categories** | | |
| | | **0–17 wks** | **>17 wks** | |
| **Females** | **130** | 86 (76.7)[1] | 44 (23.3)[1] | |
| Model A | | 0.55 (0.54, 0.56)[2] | 0.56 (0.55, 0.58)[2] | 0.1 |
| Model B[3] | | 0.55 (0.54, 0.56) | 0.56 (0.55, 0.58) | 0.1 |
| **Males** | **104** | 69 (61.1) | 35 (38.9) | |
| Model A | | 0.57 (0.56, 0.58) | 0.56 (0.54, 0.58) | 0.6 |
| Model B[4] | | 0.57 (0.56, 0.59) | 0.55 (0.54, 0.57) | 0.0917 |

Average IMT: average of means of right and left side intima media thickness (IMT).

Linear trends (P trend) were obtained in linear regression models with IMT as a continuous variable.

[1]Values are frequencies (percentages) of breastfeeding durations.

[2]Values are adjusted least squares means (95% Confidence Interval (CIs)) of IMT.

Model A adjusted for adult age at IMT measurement and the physician taking the IMT measurement.

[3]Model B among females additionally adjusted for birth year (residuals of birth year were calculated on age at IMT measurement) and maternal age at child birth.

[4]Model B among males additionally adjusted for birth year (residuals of birth year were calculated on age at IMT measurement).

with IMT among females only [38]. Such sex differences could be due to hormonal differences as well as genetics [39]. Specifically, sex chromosomes may be involved in sex variations in disease development [40]. The X chromosomes in females carries clues related to inflammation [41] and the unstable nature of the X chromosome in the outer blood cells (due to loss of the second X chromosome) is associated with autoimmune diseases as well as cardiovascular diseases in females [42, 43].

Secondly, mechanisms linking early life factors to adult IMT levels in a sex-specific way are not clear, yet female placentas have been found to be more responsive to maternal emotional distress: in female fetuses, high emotional distress of the mothers was associated with lower mRNA levels of fetal genes which prevent glucocorticoid transfer to the fetuses as well as higher mRNA levels of placental glucocorticoid receptors. In turn, in male placentas, high emotional distress was associated with high mRNA levels of genes which raise glucocorticoid inactivation providing a protective effect [44]. Animal studies have shown that high glucocorticoid exposure can result in structural alterations that can hamper heart function [45–47]. Additionally, increased maternal emotional distress was associated with high insulin like growth factor IGF2 and IGF2R mRNA levels in female but not in male placenta [44].

The strength of this study is its long follow up of participants and the fact that exposure variables were collected prospectively; careful consideration was given to the prospective assessment of potential confounders.

The study is limited by its observational nature, thus any conclusion drawn should be done with circumspection and the relatively small sample size. Attrition bias is a possibility, yet those lost to follow-up differed only slightly in their early life characteristics from those included in the analysis. Participants in the DONALD study are from a relatively high educational and socioeconomic status, so the results cannot be uncritically extrapolated to other socioeconomic (sub) populations. Data on physical activity in young adulthood were only available for subsamples, yet their consideration did not change our main findings. Nonetheless, the non-availability of behavioral variables for all participants as well as variables in other moments throughout childhood and adolescence is a limitation of our study.

## Conclusion

In conclusion, our study suggests a sex specific association between older maternal age at child birth and increased IMT in early adulthood among females.

## Supporting information

**S1 Table. Characteristics of participant's loss to follow up in adulthood.** Values are presented as means (SD), medians (IQR) or frequencies (percentage). Full breastfeeding defined as breast milk including water given to the child.
(DOCX)

**S2 Table. Association of maternal or paternal age at child birth and IMT in young adulthood.** Average IMT: average of means of right and left side intima media thickness (IMT). T: tertile, n: sample size in tertile. Linear trends (P trend) were obtained in linear regression models with IMT as a continuous variable. [1]Values are medians (25th, 75th percentiles) of early life factors. [2]Values are adjusted least squares means (95% CIs) of IMT. Model A adjusted for adult age at IMT measurement and the physician taking the IMT measurement. [3]Model B additionally adjusted for birth year (residuals of birth year were calculated on age at IMT measurement).
(DOCX)

**S3 Table. Conditional models for the association of maternal age at child birth with adult IMT in females.** Average IMT: average of means of right and left side intima media thickness (IMT). T: tertile, n: sample size in tertile. Linear trends (P trend) were obtained in linear regression models with IMT as a continuous variable. [1]Values are medians (25th, 75th percentiles) of maternal age at child birth. [2]Values are adjusted least squares means (95% CIs) of IMT. Model A adjusted for adult age at IMT measurement and the physician taking the IMT measurement. [3]Model B additionally adjusted for birth year (residuals of birth year were calculated on age at IMT measurement).
(DOCX)

**S4 Table. Association of maternal age at child birth and IMT in young adulthood among females with data on physical activity.** Average IMT: average of means of right and left side intima media thickness (IMT). T: tertile, n: sample size in tertile. Linear trends (P trend) were obtained in linear regression models with IMT as a continuous variable. [1]Values are medians (25th, 75th percentiles) of maternal age at child birth. [2]Values are adjusted least squares means (95% CIs) of IMT. Model A adjusted for adult age at IMT measurement and the physician taking the IMT measurement. [3]Model B additionally adjusted for birth year (residuals of birth year were calculated on age at IMT measurement).
(DOCX)

**S5 Table. Association of pregnancy duration or gestational weight gain with IMT in young adulthood.** Average IMT: average of means of right and left side intima media thickness (IMT). T: tertile, n: sample size in tertile. Linear trends (P trend) were obtained in linear regression models with IMT as a continuous variable. [1]p values less than 0.025 are considered significant according to Bonferroni adjustment. [2]Values are medians (25th, 75th percentiles) of early life factors. [3]Values are adjusted least squares means (95% CIs) of IMT. Model A adjusted for adult age at IMT measurement and the physician taking the IMT measurement. [4]Model B additionally adjusted for birth year (residuals of birth year were calculated on age at IMT measurement).
(DOCX)

**S6 Table. Association of birthweight for gestational age or birthweight and adult IMT.**
AGA: appropriate for gestational age, LGA: large for gestational age, SGA: small for gestational age. AGA, LGA and SGA defined according to German sex-specific birth weight and length-for-gestational-age curves. Linear trends (P difference) were obtained in linear regression models with IMT as a continuous variable and birthweight according to gestational age as a 3 level categorical variable (0 = SGA; 1 = AGA; 2 = LGA with AGA set as the reference category in the models). [1]p difference and trend less than 0.025 are considered significant according to Bonferroni adjustment. [2]Values are frequencies (percentages) of birthweight according to gestational age. [3]Values are adjusted least squares means (95% CIs) of IMT. [4]Values are medians (25th, 75th percentiles) of birthweight. Model A adjusted for adult age at IMT measurement and the physician taking the IMT measurement. Model B additionally adjusted for birth year (residuals of birth year were calculated on age at IMT measurement).
(DOCX)

## Author Contributions

**Conceptualization:** Thomas Remer, Anette E. Buyken.

**Data curation:** Katharina Penczynski.

**Formal analysis:** Juliana Nyasordzi.

**Project administration:** Thomas Remer, Anette E. Buyken.

**Supervision:** Anette E. Buyken.

**Writing – original draft:** Juliana Nyasordzi.

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
