## [Decision Letter · Decision Letter 0]

20 Jan 2020

PONE-D-19-30815

Early life factors and their relevance to intima-media thickness of the common carotid artery in early adulthood

PLOS ONE

Dear Dr. Buyken,

Thank you for submitting your manuscript to PLOS ONE. After careful consideration, we feel that it has merit but does not fully meet PLOS ONE’s publication criteria as it currently stands. Therefore, we invite you to submit a revised version of the manuscript that addresses the points raised during the review process.

As you will recognize from the comments of the reviewers major points of critique were raised especially regarding study design and methodology as well as presentation of results.

We would appreciate receiving your revised manuscript within 2 months. To enhance the reproducibility of your results, we recommend that if applicable you deposit your laboratory protocols in protocols.io, where a protocol can be assigned its own identifier (DOI) such that it can be cited independently in the future. For instructions see: http://journals.plos.org/plosone/s/submission-guidelines#loc-laboratory-protocols

We look forward to receiving your revised manuscript.

Kind regards,

Rudolf Kirchmair

Academic Editor

PLOS ONE

Journal Requirements:

The DONALD Study is supported by the Ministry of Innovation, Science,

Research and Technology of the State of North Rhine-Westphalia, Germany. Juliana Nyasordzi is

a PhD candidate co-sponsored by the government of Ghana (Ministry of Education) and the

German government (Deutscher Akademischer Austauschdienst German Academic Exchange

Service (DAAD).

3. We noticed you have some minor occurrence(s) of overlapping text with the following previous publication(s), which needs to be addressed:

https://doi.org/10.3390/nu10040488

https://doi.org/10.1038/oby.2007.57

https://doi.org/10.1016/j.atherosclerosis.2014.09.027

https://doi.org/10.1007/s00394-016-1286-x

https://doi.org/10.1093/ajcn/84.6.1449

https://doi.org/10.3945/ajcn.2009.28259

In your revision ensure you cite all your sources (including your own works), and quote or rephrase any duplicated text outside the Methods section. Further consideration is dependent on these concerns being addressed.

Reviewers' comments:

Reviewer's Responses to Questions

**Comments to the Author**

1. Is the manuscript technically sound, and do the data support the conclusions?

Reviewer #1: Partly

Reviewer #2: Yes

2. Has the statistical analysis been performed appropriately and rigorously? 

Reviewer #1: No

Reviewer #2: Yes

3. Have the authors made all data underlying the findings in their manuscript fully available?

Reviewer #1: No

Reviewer #2: Yes

4. Is the manuscript presented in an intelligible fashion and written in standard English?

Reviewer #1: Yes

Reviewer #2: Yes

5. Review Comments to the Author

Reviewer #1: This was a long follow-up study on early life factors and their relevance to early adulthood IMT German participants. None of the early life factors was associated with IMT except the maternal age only among females. The manuscript gives insight on a major public health problem; however, some concerns exist when interpreting the results.

1- It is not clear what the aim of study was. Whether constructing a prediction model of IMT based on early life factors or investigating the association of a specific factor with IMT. These two approaches are totally different in the statistical analysis. The manuscript should be unified based on its objective.

2- Too many factors were investigated with IMT. Multiple testing (multiplicity) is a common phenomenon and would lead to probability of false positive results. How the authors considered this event?

3- Strong evidence has been shown that one of the important factors determining the IMT is physical activity. The current research lacks measuring physical activity. Those with higher IMT may have lower physical activity. And we cannot relate the higher IMT to only maternal age.

4- Nothing has been mentioned on the characteristics of loss to follow-up participants. The high rate of loss to follow-up may confound the results.

5- The mean/median follow-up time was not stated. As authors declared that inclusion of participants in the cohort and IMT measurement were conducted both in different ages of participants, it more logical to use the survival analysis and include censoring time.

6- Birth year really does not have any added value. The range of the inclusion of participants was less than a decade. Unless investigators believe that something had happened during the recruitment years in the situation.

7- Blood pressure and body mass index have main effect on atherosclerosis in particular carotid IMT. Were these factors considered during analysis or not?

8- Does testing the interaction of sex and early life factors on early adulthood IMT sound biologically? The discussion actually did not explain this main finding from the early life point of view. The sex difference finding of studies in adult age was mentioned. What are the probable mechanisms for the sex difference effect of early life factors such as maternal age?

9- It is not clear why adulthood waist circumference was adjusted. There are many other important factors in adulthood, which affect adulthood IMT.

10- For non-significant findings, it is not necessary to discuss the effect of early life factors on IMT. More importantly, the explanation of non-significant results is required.

11- To test the trend of baseline characteristics, analysis of variance is not appropriate. It is mainly shows the difference and is used for comparison.

Reviewer #2: Dear Editor and Authors,

thank you for the opportunity to review this nice manuscript. I have some comments in order to improve the manuscript, but in general perspective, the manuscript brings relevant data about the issue.

Introduction: In general, the section is well organized and provides all the relevant details that the readers need to understand the background leading the authors to perform this research. My only comment is the absence of any data about other relevant behaviors / factors during adolescence (also a sensitive period to development of cardiovascular diseases). I will bring this aspect into the light later in the discussion section as well.

Methods

The section is well described. I do not have many comments. I only comment is to insert a flowchart describing all variables considered, as well as the the time point where those variables were collected.

Results

The section is particularly confusing. I suggest a different approach to present the results.

Table 1: it would be nice to compare all variables according to sex, mainly because the findings seems sex-dependent.

The comparison according to IMT tertile seems a weak stratification (there is no data about the number of participants with altered values for IMT). Even a few number, split out the sample according altered and normal IMT value seems more appropriate.

Is there any relationship between maternal age and the other early life variables considered (eg. low birthweight)? Due to the relevance of maternal age in your findings, a table describing that would welcome.

Discussion: The section need to the reduced in size. There was only one significant association, but there are four pages of discussion. Please, focus only on significant findings. Moreover, the ways the authors built the sections leads the reader to believe that behavioral variables in other moments of life are not relevant into this phenomenon. For instance, IMT in adulthood is affected by sports participation during adolescence (Werneck et al. 2018). Please, I would suggest to highlight the relevance of these data and also its absence in the models created.

References

Werneck AO, Lima MCS, Agostinete RR, Silva DR, Turi-Lynch BC, Codogno JS, Fernandes RA. Association between Sports Participation in Early Life and Arterial Intima-Media Thickness among Adults. Medicina (Kaunas). 2018;54(5). pii: E85. doi: 10.3390/medicina54050085.

6. PLOS authors have the option to publish the peer review history of their article (what does this mean?). If published, this will include your full peer review and any attached files.

Reviewer #1: No

Reviewer #2: No

---

## [Author Response · Author response to Decision Letter 0]

26 Mar 2020

POINT-BY-POINT RESPONSE TO EDITORIAL BOARD MEMBER’S AND REVIEWERS COMMENTS 

Manuscript: [PONE-D-19-30815] - [EMID:15f54e8fc6e19a2a]

Title of paper: Early life factors and their relevance to intima-media thickness of the common carotid artery in early adulthood.

Authors: Juliana Nyasordzi, Katharina Penczynski, Thomas Remer, Anette E. 

Buyken.

General Comments

We cherish the suggestions made by the editor and reviewers all in an attempt to improve the quality of the manuscript. We have carefully considered the concerns raised by the editor and reviewers and amended the manuscript appropriately. 

Editor’s Query 1 

Response to EQ1

We have ensured that the manuscript meets PLOS ONE’S style requirements.

Editor’s Query 2

Response to EQ2

The funding information has been removed from the manuscript and is reported in the funding statement section of the online submission form as follows: The DONALD Study is supported by the Ministry of Innovation, Science, Research and Technology of the State of North Rhine-Westphalia, Germany. Juliana Nyasordzi is a PhD candidate co-sponsored by the government of Ghana (Ministry of Education) and the German government (Deutscher Akademischer Austausch-dienst German Academic Exchange Service (DAAD). 

The authors declare no conflict of interest. The funders have no role in the design of the study, collection, analyses and interpretation of the data, the writing of the manuscript, and in the decision to submit the findings for publication.

Editor’s Query 3

We noticed you have some minor occurrence(s) of overlapping text with the following previous publication(s), which needs to be addressed:

https://doi.org/10.3390/nu10040488

https://doi.org/10.1038/oby.2007.57

https://doi.org/10.1016/j.atherosclerosis.2014.09.027

https://doi.org/10.1007/s00394-016-1286-x

https://doi.org/10.1093/ajcn/84.6.1449

https://doi.org/10.3945/ajcn.2009.28259

In your revision ensure you cite all your sources (including your own works), and quote or rephrase any duplicated text outside the Methods section. Further consideration is dependent on these concerns being addressed.

Response to EQ3

The overlapping texts have been reworded accordingly and duly cited.

This has resulted in a complete rewording of paragraphs � see lines 30-31, 39-40, 49-56, 57-64, 85-87, 92-95, 98-105, 112-114, 116-117, 119, 122, 134-143, 169-173 in the manuscript. 

In addition, multiple minor changes of the text were performed throughout the manuscript. 

All changes are highlighted in the marked up copy in red.

Editor’s Query 4

We note that you have indicated that data from this study are available upon request. PLOS only allows data to be available upon request if there are legal or ethical restrictions on sharing data publicly. For information on unacceptable data access restrictions, please see http://journals.plos.org/plosone/s/data-availability#loc-unacceptable-data-access-restrictions.

Response to EQ4

 In response to the General data protection regulation issued by the European Union, the official data policy has been adapted by the DONALD study group including the decision that datasets can generally only be obtained upon request. The main reason for this being that the study is ongoing (i.e. data cannot be anonymized) and due to its comparatively small sample size there is a higher risk for identification of persons even in pseudonomized data sets. This regulation has been developed as specified by the data protection officer of the University of Bonn Dr. Jörg Hartmann.

Our text dealing with this now reads:

Data from this study are available on request. The DONALD Study is still ongoing and the comparably small sample requires specific precautions to avoid potentially identifying participant information. This has been imposed by the data protection officer of the University of Bonn as an official ethical restriction. Requests for data access may be sent to the local data protection coordinator Heinz Rinke, email: rinke@uni-bonn.de at the DONALD Study Dortmund of the University of Bonn. 

Reviewers’ comments to the Author:

Reviewer #1 (Remarks to the Author):

Query 1

It is not clear what the aim of study was. Whether constructing a prediction model of IMT based on early life factors or investigating the association of a specific factor with IMT. These two approaches are totally different in the statistical analysis. The manuscript should be unified based on its objective.

Response 1

We apologize for not having been clear with respect to our aims. We were not interested in constructing a prediction model of IMT. Instead, our aim was to assess the relevance of specific early life factors for adult intima media thickness (IMT), a surrogate of cardiovascular disease (CVD). Our hypothesis was that it is plausible that a range of early life factors could be associated with CVD. Specifically, we had four domains of hypotheses which we tested for,

1. Early life factors related to indicators of intrauterine growth (i.e. birth weight, birth weight-for-gestational-age (SGA, AGA and LGA) and their association with IMT. 

2. Early life factors related to pregnancy (i.e. pregnancy duration and gestational weight gain) and their association with IMT. 

3. Early life factors related to parental age at child birth (i.e. maternal and paternal age) and their association with IMT.

4. Postnatal nutrition (i.e. breastfeeding) and its association with IMT.

The aim of the study in the manuscript has been reworded to reflect these four domains of hypotheses tested. Please see lines 69-79 in the manuscript. 

Query 2

Too many factors were investigated with IMT. Multiple testing (multiplicity) is a common phenomenon and would lead to probability of false positive results. How the authors considered this event?

Response 2

We agree that multiple testing is a concern, enhancing the probability of false positive results. 

As outlined above, we did however address four different sets of hypotheses, i.e. for each of these different mechanisms have been proposed. 

For the first (intrauterine growth) and the second (pregnancy) hypotheses we do however, concur that their relevance was assessed by two variables each, i.e. the intrauterine growth hypothesis is addressed by birthweight and birthweight according to gestational age (please, note that SGA, AGA, LGA represent one 3 level categorical variable)) and the pregnancy hypothesis was addressed by both pregnancy duration and gestational weight gain. We therefore now apply Bonferroni correction for these two hypotheses in that we include a footnote explaining that the respective results are considered significant if p<0.025.

In our view, maternal age at birth and paternal age at birth address two separate potential mechanisms, hence no Bonferroni adjustment was applied to the variables reflecting this hypothesis. 

 Please see line 542 Table S5 and line 553 Table S6 in the supplementary data section. 

Query 3

Strong evidence has been shown that one of the important factors determining the IMT is physical activity. The current research lacks measuring physical activity. Those with higher IMT may have lower physical activity. And we cannot relate the higher IMT to only maternal age.

Response 3

We reckon that other factors such as physical activity can influence the IMT. Unfortunately, assessment of physical activity was incorporated into the DONALD study protocol in 2004 for participation in sport and 2012 for estimated energy expenditure during participation in sport. Hence information on participation in sport in young adulthood is only available in a subsample of n=214 (n=123 female and n=91 male) and information on estimated energy expenditure during sport in young adulthood only in a subsample of n=123 (n=68 female and n=55 males).

Note that participation in sport refers to both organized and unorganized sports, similarly, estimated energy expenditure is based on information acquired for both organized and unorganized sport. 

Please, find below the results from the two sensitivity analyses in the respective samples, which reflect that our main finding (association of maternal age at birth with IMT) was retained in the smaller samples and not affected by additional consideration of participation in sport or estimated energy expenditure during participation in sport. Please, note that the change in the ß for maternal age at birth was < 1% upon additional consideration of participation in sport (comparing model B without vs model B with additional consideration of sport) and ˂ 4% upon additional consideration of estimated energy expenditure during participation in sport (comparing model B without vs model B with additional consideration of energy expenditure during sport). 

S4 Table. Association of maternal age at child birth and IMT in young adulthood among females with data on physical activity 

 Average IMT (mm) 

 N P trend

Females 123 T1 (n=32) T2 (n=46) T3 (n=45) 

Maternal age at child birth (yrs)1 27 (25, 28) 30 (29, 31) 34 (33, 36) 

Model A2 0.54 (0.53, 0.56) 0.56 (0.54, 0.57) 0.57 (0.56, 0.58) 0.0017

Model B3 0.55 (0.53, 0.56) 0.56 (0.54, 0.57) 0.57 (0.56, 0.58) 0.0016

Model B including sport 0.54 (0.53, 0.56) 0.56 (0.54, 0.57) 0.57 (0.56, 0.58) 0.0014

Females 68 T1 (n=26) T2 (n=18) T3 (n=24) 

Maternal age at child birth (yrs)1 28 (27, 29) 31 (30, 32) 36 (34, 37) 

Model A2 0.55 (0.53, 0.57) 0.56 (0.54, 0.58) 0.58 (0.56, 0.60) 0.0144

Model B3 0.55 (0.53, 0.57) 0.56 (0.54, 0.58) 0.58 (0.56, 0.60) 0.0092

Model B including energy expenditure during sport 0.55 (0.53, 0.57) 0.56 (0.54, 0.58) 0.58 (0.56, 0.60) 0.0095

Average IMT: average of means of right and left side intima media thickness (IMT) 

 T: tertile, n: sample size in tertile. 

Linear trends (P trend) were obtained in linear regression models with IMT as a continuous variable. 

1Values are medians (25th, 75th percentiles) of maternal age at child birth. 

2Values are adjusted least squares means (95% CIs) of IMT. Model A adjusted for adult age at IMT measurement and the physician taking the IMT measurement.

3Model B additionally adjusted for birth year (residuals of birth year were calculated on age at IMT measurement).

The results of this sensitivity analysis have been included as additional supplementary material in the manuscript (Table S4 line 532).

Additional sensitivity analyses were performed in subsamples of males, adjusting for participation in sport (n=91) and estimated energy expenditure during participation in sport (n=55), so as to investigate whether the trend association of breastfeeding with IMT was retained. However, no trends were observed in these smaller subsamples irrespective of additional consideration of sport or energy expenditure.

In view of the fact that this change in association is clearly attributable to the reduction in sample size (and not to the consideration of physical activity variables in adulthood), we refrain from including the results of this sensitivity analyses as additional supplementary material. 

The results of the sensitivity analyses are however reported in the paper. Please see lines 269-272.

 

Query 4

Nothing has been mentioned on the characteristics of loss to follow-up participants. The high rate of loss to follow-up may confound the results. 

Response 4

We agree that loss to follow-up may result in attrition bias. Please, note some specifics arising for the DONALD Study:

• The children who were initially recruited for the DONALD Study differed con-siderably in age and prospectively collected data on breastfeeding was not always available.

• Due to the open cohort design, many DONALD participants had not yet reached young adulthood by the time of this analysis. 

• Follow-up into adulthood was not planned at the inception of the study. Partici-pants are invited since 2004 onwards, to return for further visits at ages 18, 21, 25, 30, 35 etc. However, not all participants followed this invitation. 

To address your query, we assessed the characteristics (see table S1) provided by the par-ticipants at inclusion in the study among those who 

- had provided data on birth year, birth weight, gestational duration, and maternal age at birth 

- had reached the age of 18 years by 2014

- yet had not returned for an IMT measurement in young adulthood. 

This comparison reveals that there are minor differences between the sample included in our analysis and those lost to follow up (see Table 1 in the main paper for comparison): mothers were younger at birth i.e. 29.4 and 29.8 years for male and female offspring ver-sus 30.6 and 30.3 years, children were born earlier, i.e. 1987 in both sexes versus 1989 and 1988, respectively and fewer offspring were breastfed for a long duration, i.e. 31% and 27% among male and female offspring in the lost-to follow-up sample versus 34% among both sexes in the analyzed sample.

This information has been included in the manuscript, please see lines 224-227.

In addition, the specifics of the DONALD study and the selection of the sample as out-lined above are now reported together in one paragraph, so as to make this information more easily understandable for the reader � see lines 98-103.

Taken together, in spite of the minor differences attrition bias remains a possibility and this is now acknowledged in the limitations � see lines 342-344.

 

Table S1. Early life characteristics of participants lost to follow up

Variables Males Females 

Early life factors N N 

Maternal age at child birth (yrs) 157 29.4 (4.2) 177 29.8 (4.0)

Paternal age at child birth (yrs) 145 32.7 (5.1) 159 33.0 (6.2)

Pregnancy duration (wks) 157 40 (40, 41) 177 40 (39, 41)

Gestational weight gain (kg) 150 13.2 (4.4) 171 12.9 (4.2)

Birthweight (g) 157 3561 (476) 177 3445 (417)

Full breastfeeding 135 144 

 Never (0-2 weeks) 53 (39.3%) 45 (31.3%)

 Short duration (3-17 weeks) 40 (29.6%) 60 (41.7%)

 Long duration (>17 weeks) 42 (31.1%) 39 (27.0%)

Birth year 157 1987 (1982, 1990) 177 1987 (1983, 1991)

 Values are presented as means (SD), medians (IQR) or frequencies (percentage).

 Full breastfeeding defined as breast milk including water given to the child.

Query 5

The mean/median follow-up time was not stated. As authors declared that inclusion of participants in the cohort and IMT measurement were conducted both in different ages of participants, it more logical to use the survival analysis and include censoring time.

Response 5

We apologize for the confusion, which may have arisen from the fact that the open-cohort design of our study is very uncommon. The study is ongoing and participants are still being recruited onto the study each year. However, those participants that were born until 1996 could have been included in this analysis, since they reached the age of 18 by 2014. Hence, for this analysis the mean follow-up is equivalent with the mean age at IMT measurement, which was 23 years in males and 24 years in females. 

This has been stated in the methods of the manuscript � see lines 104-105.

Please, note that our outcome was a continuous variable (IMT), representing a surrogate marker of CVD risk, i.e. we were not following them up for a disease outcome per se in which case survival analysis and censoring time would have been appropriate. In addition, as explained above age at IMT measurement is largely attributable to the changes in the study design (invitation for adult visits since 2004, invitation to IMT measurements since 2008), i.e. censoring time is not related to the outcome. Hence use of survival analysis appears inappropriate, which is why we performed a multiple linear regression analysis. Please, note that we adjusted for age at IMT measurement.

 

Query 6

Birth year really does not have any added value. The range of the inclusion of participants was less than a decade. Unless investigators believe that something had happened during the recruitment years in the situation.

Response 6

We politely disagree. The data presented for birth year in table 1 are interquartile ranges (as indicated in the footnote). As explained above due to the open-cohort design of the DONALD study, birth year ranged from 1985-1996, i.e. slightly more than a decade. 

Selection of birth year as a potential confounder emerged from our strict strategy of confounder selection outlined in the method section. To our surprise, birth year was identified as a relevant confounder hence its inclusion in the model. Removal of this variable results in stronger associations between the early life factors and IMT, i.e. we believe that it would not be correct to report the stronger associations, knowing that they may be confounded by birth year.

Query 7

Blood pressure and body mass index have main effect on atherosclerosis in particular carotid IMT. Were these factors considered during analysis or not?

Response 7

We agree that – similar to waist circumference – BMI or blood pressure may act as a factor explaining the link between early life factors and adult IMT. We hence carried out additional sensitivity analyses adding either BMI or systolic or diastolic blood pressure (in a slightly smaller sample) in adulthood in the respective conditional models for early life factors that were found to be significantly associated with adult IMT. As illustrated below, the association between maternal age at birth and IMT in young adulthood was retained in all three additional conditional models. Please, note that the change in ß for maternal age at birth was <3% in the conditional models (when compared to model B among n=142) with systolic or diastolic blood pressure and <1% in conditional models with BMI (when compared to model B among n=140). 

S3 Table. Conditional models for the association of maternal age at child birth with adult IMT in females

 Average IMT (mm) 

 N P trend

Females 142 T1 (n=44) T2 (n=50) T3 (n=48) 

Maternal age at child birth (yrs) 1 27 (25, 28) 30 (29, 31) 34 (33, 36) 

Model A2 0.54 (0.53, 0.56) 0.55 (0.54, 0.57) 0.57 (0.56, 0.58) 0.0010

Model B3 0.54 (0.53, 0.56) 0.55 (0.54, 0.56) 0.57 (0.55, 0.58) 0.0025

Conditional model including waist circumference 0.54 (0.53, 0.56) 0.55 (0.54, 0.56) 0.57 (0.56, 0.58) 0.0025

Conditional model including BMI 0.54 (0.53, 0.56) 0.55 (0.54, 0.56) 0.57 (0.56, 0.58) 0.0025

 140 T1 (n=44) T2 (n=50) T3 (n=46) 

Maternal age at child birth (yrs) 1 27 (25, 28) 30 (29, 31) 34 (33, 36) 

Model A2 0.54 (0.53, 0.56) 0.55 (0.54, 0.57) 0.57 (0.55, 0.58) 0.0012

Model B3 0.54 (0.53, 0.56) 0.55 (0.54, 0.56) 0.57 (0.55, 0.58) 0.0029

Conditional model including systolic blood pressure 0.55 (0.53, 0.56) 0.55 (0.54, 0.56) 0.57 (0.55, 0.58) 0.0036

Conditional model including diastolic blood pressure 0.54 (0.53, 0.56) 0.55 (0.54, 0.56) 0.57 (0.55, 0.58) 0.0024

Average IMT: average of means of right and left side intima media thickness (IMT). 

 T: tertile, n: sample size in tertile. 

Linear trends (P trend) were obtained in linear regression models with IMT as a continuous variable. 

1Values are medians (25th, 75th percentiles) of maternal age at child birth. 

2Values are adjusted least squares means (95% CIs) of IMT. Model A adjusted for adult age at IMT measurement and the physician taking the IMT measurement.

 3Model B additionally adjusted for birth year (residuals of birth year were calculated on age at IMT measurement). 

Since none of the conditional models affected the main result, the conditional model for waist circumference was removed from tables 2 and S2, S5-S6 and all conditional models are reported in Appendix table S3 lines 522 and the text only � lines 244-246 and lines 266-268.

Query 8

Does testing the interaction of sex and early life factors on early adulthood IMT sound biologically? The discussion actually did not explain this main finding from the early life point of view. The sex difference finding of studies in adult age was mentioned. What are the probable mechanisms for the sex difference effect of early life factors such as maternal age?

Response 8 

We apologize for not having separated observations in adulthood more clearly from potential mechanisms acting in early life. We now separately appraise that 

1.) There is a considerable body of evidence that biological factors are responsible for sex dif-ferences in chronic diseases (Kautzky-Willer, 2016) (Kuznetsova, 2018) (Den Ruijter et al., 2015) and potential genetic differences are discussed in this context � lines 317-327.

2.) Epigenetic mechanisms and early nutritional factors and psychosocial stress acting in early life may contribute to sex-differences in adult IMT. Specifically, we discuss

• maternal emotional distress during pregnancy, which can result in adverse changes in placental gene expression, in female fetuses only. 

• susceptibility in female animal fetuses to higher transfer of bioactive glucocorticoids that can impair cardiac structure and function 

lines 328-337.

Query 9

 It is not clear why adulthood waist circumference was adjusted. There are many other important factors in adulthood, which affect adulthood IMT. 

Response 9

Waist circumference is a measure of central adiposity and is a useful predictor of atherosclerosis (Knowles et al., 2010). It is hence plausible, that waist circumference may act as a link between early life factors and adult IMT – similar to BMI and blood pressure, please see query 7. This is why we had included waist circumference in a conditional model.

As outlined above, neither inclusion of adult waist circumference, nor BMI, nor systolic or diastolic blood pressure affected the significant association between maternal age at birth and adult IMT. To avoid presenting a large number of non-informative tables, we removed all conditional models from the tables. Instead, we now include a statement in the results, that additional consideration of these variables did not explain our finding. All conditional models are presented in table S3 in the appendix of the manuscript.

Query 10

For non-significant findings, it is not necessary to discuss the effect of early life factors on IMT. More importantly, the explanation of non-significant results is required.

Response 10

Agreed. We substantially reduced our discussion on the non-significant findings, moving some of the content to the introduction to explain our lines of hypotheses. 

Query 11

To test the trend of baseline characteristics, analysis of variance is not appropriate. It is mainly shows the difference and is used for comparison.

Response 11

Agreed. We removed all tests in the baseline characteristics.

Reviewer #2 (Remarks to the Author):

Reviewers’ comments to the Author:

Query 12

My only comment is the absence of any data about other relevant behaviors / factors during adolescence (also a sensitive period to development of cardiovascular diseases). I will bring this aspect into the light later in the discussion section as well.

Response 12

We acknowledge that physical activity can influence the IMT. However, assessment of physical activity was included in the DONALD study protocol in 2004 for participation in sport and 2012 for estimated energy expenditure during participation in sport, thus information on participation in sport is only available for a subsample of n=123 female and n=91 male and information on estimated energy expenditure during sport in young adulthood only for n=68 female and n=55 males.

We performed a sensitivity analysis in the subsamples with information on participation in sport or estimated energy expenditure during participation in sport. The results indicate the association of maternal age at birth with IMT was retained and not affected by additional consideration of physical activity. Please see response to query 3.

 

Query 13

Insert a flowchart in the methods describing all variables considered, as well as the time point where those variables were collected

Response 13

A flowchart of IMT measurement has been inserted in the methods section� lines 117-119.

In terms of the time points, we apologize for not having been clear. 

• Early life factors referring to gestation and birth were inquired at the first visit to the study (usually at ages 0.25 or 0.5 years)

• Data on full breastfeeding (not given solid foods and no liquids daily except breast milk, tea, or water) were inquired at 3 or 6 months and prospectively at subsequent visits until weaning.

• Data on IMT were measured during young adulthood at ages 18, 21, 25, 30, 35 and 40 years

This information has been added to the manuscript � lines 99-100.

Query 14

I suggest a different approach to present the results.

Table 1: compare all variables according to sex, mainly because the findings seems sex-dependent.

Response 14

Thanks for your suggestion, Table 1 has been revised and the results are presented according to sex.

Query 15

Is there any relationship between maternal age and the other early life variables considered (eg. low birthweight)? Due to the relevance of maternal age in your findings, a table describing that would welcome.

Response 15

Maternal age at birth was related to paternal age at birth (p˂0.0001) and pregnancy duration (p=0.03) but not to birth weight according to gestational age (i.e. SGA, AGA, LGA), gestational weight gain, birthweight or breastfeeding (p>0.05). 

Following our strict procedure for confounder selection, paternal age at birth or pregnancy duration did not emerge as confounders for models examining the relevance of maternal age at birth for adult IMT. Please, note that despite its high correlation with maternal age at birth, there was no collinearity with maternal age at birth (i.e. tolerances of the model including both variables were acceptable).

Since the fact that paternal age at birth did not affect the association may have implication for the interpretation of the results, this was stressed in the results � see lines 241 and 244 and referred to in the discussion � see lines 298-300.

We would like to abstain from presenting a table on the relation between maternal age at birth with other early life factors for the following reasons:

• Maternal age at birth emerged as relevant for IMT during analyses, i.e. our aim was not to describe the association between maternal age at birth and other early life factors.

• The other early life factors were not of relevance for the main association that emerged during analyses, i.e. the relation of maternal age with adult IMT.

Should the Editor feel that the inclusion of such a table in the Appendix would be of interest, we are however prepared to do so.

Query 16

The discussion needs to be reduced and focused only on significant findings. Moreover, the ways the authors built the sections leads the reader to believe that behavioral variables in other moments of life are not relevant into this phenomenon. For instance, IMT in adulthood is affected by sports participation during adolescence (Werneck et al. 2018). Please, I would suggest to highlight the relevance of these data and also its absence in the models created.

Response 16

Agreed. We substantially reduced our discussion on the non-significant findings, moving some of the content to the introduction to explain our lines of hypotheses. �see response to queries 1 & 10, reviewer 1.

The reviewer correctly points out that other behavioral variables such as physical activity are relevant for adult IMT. We expanded our analysis carrying out a sensitivity analysis including participation in physical activity as an additional covariate but this did not change the result that maternal age at child birth is relevant for IMT in early adulthood in females but not in males. � see response to query 3, reviewer 1.

We have added the non-availability of behavioral variables in other moments of life that are relevant to IMT as a limitation of our study in the discussion. � see lines 347-349.

 References

Den Ruijter HM, Haitjema S, Asselbergs FW, Pasterkamp G. Sex matters to the heart: A special issue dedicated to the impact of sex related differences of cardiovascular diseases. Atherosclerosis. 2015;241:205–7. doi:10.1016/j.atherosclerosis.2015.05.003.

Kautzky-Willer A, Harreiter J, Pacini G. Sex and Gender Differences in Risk, Pathophysiology and Complications of Type 2 Diabetes Mellitus. Endocr Rev. 2016; 37:278–316. doi:10.1210/er.2015-1137.

Knowles KM, Paiva LL, Sanchez SE, Revilla L, Lopez T, Yasuda MB, et al. Waist Circumference, Body Mass Index, and Other Measures of Adiposity in Predicting Cardiovascular Disease Risk Factors among Peruvian Adults. Int J Hypertens. 2011; 2011:931402. doi:10.4061/2011/931402.

Kuznetsova T. Sex Differences in Epidemiology of Cardiac and Vascular Disease. Adv Exp Med Biol. 2018;1065:61–70. doi:10.1007/978-3-319-77932-4_4.

---

## [Decision Letter · Decision Letter 1]

1 May 2020

Early life factors and their relevance to intima-media thickness of the common carotid artery in early adulthood

PONE-D-19-30815R1

Dear Dr. Buyken,

We are pleased to inform you that your manuscript has been judged scientifically suitable for publication and will be formally accepted for publication once it complies with all outstanding technical requirements.

With kind regards,

Rudolf Kirchmair

Academic Editor

PLOS ONE

Additional Editor Comments (optional):

Reviewers' comments:

Reviewer's Responses to Questions

**Comments to the Author**

1. If the authors have adequately addressed your comments raised in a previous round of review and you feel that this manuscript is now acceptable for publication, you may indicate that here to bypass the “Comments to the Author” section, enter your conflict of interest statement in the “Confidential to Editor” section, and submit your "Accept" recommendation.

Reviewer #1: All comments have been addressed

Reviewer #2: All comments have been addressed

2. Is the manuscript technically sound, and do the data support the conclusions?

Reviewer #1: Yes

Reviewer #2: Yes

3. Has the statistical analysis been performed appropriately and rigorously? 

Reviewer #1: Yes

Reviewer #2: Yes

4. Have the authors made all data underlying the findings in their manuscript fully available?

Reviewer #1: Yes

Reviewer #2: Yes

5. Is the manuscript presented in an intelligible fashion and written in standard English?

Reviewer #1: Yes

Reviewer #2: No

6. Review Comments to the Author

Reviewer #1: All the comments have been fully addressed and responded. The important restrictions have been added to limitation part.

Reviewer #2: Comparing with the previous version, all my comments were properly addressed by the author. Therefore, there are no further comments.

7. PLOS authors have the option to publish the peer review history of their article (what does this mean?). If published, this will include your full peer review and any attached files.

Reviewer #1: No

Reviewer #2: No

---

## [Editor Report · Acceptance letter]

5 May 2020

PONE-D-19-30815R1 

Early life factors and their relevance to intima-media thickness of the common carotid artery in early adulthood 

Dear Dr. Buyken:

I am pleased to inform you that your manuscript has been deemed suitable for publication in PLOS ONE. Congratulations! Your manuscript is now with our production department. 

With kind regards,

on behalf of

Prof Rudolf Kirchmair 

Academic Editor

PLOS ONE